# The CpG Island Methylator Phenotype Status in Synchronous and Solitary Primary Colorectal Cancers: Prognosis and Effective Therapeutic Drug Prediction

**DOI:** 10.3390/ijms25105243

**Published:** 2024-05-11

**Authors:** Yun-Yun Weng, Ming-Yii Huang

**Affiliations:** 1Department of Radiation Oncology, Kaohsiung Medical University Hospital, Kaohsiung Medical University, Kaohsiung 807, Taiwan; yunyunweng.kate@gmail.com; 2Department of Radiation Oncology, School of Medicine, College of Medicine, Kaohsiung Medical University, Kaohsiung 807, Taiwan

**Keywords:** CpG island methylator phenotype, synchronous colorectal cancers, therapeutic drug

## Abstract

Synchronous colorectal cancer (sCRC) is characterized by the occurrence of more than one tumor within six months of detecting the first tumor. Evidence suggests that sCRC might be more common in the serrated neoplasia pathway, marked by the CpG island methylator phenotype (CIMP), than in the chromosomal instability pathway (CIN). An increasing number of studies propose that CIMP could serve as a potential epigenetic predictor or prognostic biomarker of sCRC. Therapeutic drugs already used for treating CIMP-positive colorectal cancers (CRCs) are reviewed and drug selections for sCRC patients are discussed.

## 1. Introduction

Colorectal cancer (CRC) was the third most common cancer and the third leading cause of cancer-related deaths in Taiwan in 2021 [1]. Globally, it ranked as the third most common and the second leading cause of cancer-related deaths in 2020 [2]. CRCs are highly heterogeneous in molecular carcinogenesis [3,4]. Numerous treatment choices corresponding to each phenotype have become available in recent years.

Solitary primary colorectal cancer (spCRC) is defined as a primary colorectal carcinoma detected during staging. The term “synchronous colorectal cancer” (sCRC) is used to describe the identification of more than one primary colorectal carcinoma in a patient at the time of diagnosis. Approximately 1.1% to 8.1% of all CRCs are sCRCs [5]. This review article aimed to investigate the molecular features that might play a role in therapeutic decisions in treating sCRC. The candidate markers included microsatellite instability (MSI) status, DNA mismatch repair (MMR) protein expression, CpG island methylator phenotype (CIMP) status, Kirsten rat sarcoma viral oncogene homologue (*KRAS*), neuroblastoma rat sarcoma viral oncogene homologue (*NRAS*), and B-Raf proto-oncogene serine/threonine kinase (*BRAF*) gene mutations.

Evidence suggests that sCRC might be more common in the serrated neoplasia pathway, marked by the CIMP, than in the chromosomal instability pathway (CIN). An increasing number of studies propose that CIMP could serve as a potential epigenetic predictor or prognostic biomarker of sCRC. Therapeutic drugs already used for treating CIMP-positive CRCs are reviewed and drug selections for sCRC patients are discussed.

### 1.1. Clinical Characteristics of Synchronous Colorectal Cancer Compared to Solitary Primary Colorectal Cancer

The Warren and Gates criteria were developed to provide a clinical framework for diagnosing sCRC by assessing histopathological features evident in CRC biopsies [6]. These criteria are widely employed and encompass the following essential components: (1) each neoplasm must manifest definite malignant characteristics; (2) each lesion must exhibit distinctiveness; (3) the likelihood of one lesion representing a metastatic spread from the other must be definitively ruled out; and (4) synchronous lesions must be diagnosed concurrently or within a maximum interval of 6 months from the initial identification [7]. The tumors are at least 4 cm apart in normal colorectal mucosa and exclusive of submucosal spread or satellite lesions [8]. The most invasive lesion (with the highest pT) is identified as the primary “index” tumor as the reference for the pathological and molecular classification of sCRC [9].

A systematic search of the PubMed database covering the period from 1981 to 2013 was conducted to identify relevant research articles pertaining to sCRCs [5]. The prevalence of sCRCs exhibited variation, ranging from 1.1% to 8.1% across diverse geographic regions, including Europe, Asia, and America. Specifically, prevalence rates reported in Europe ranged from 1.1% to 8.1%, in Asia from 1.1% to 8.1%, and in America from 1.2% to 7.0% [5]. Regarding potential differences in age range between sCRC and spCRC occurrences, the medical community has yet to reach a consensus. The mean ages at presentation for sCRC have ranged from 47 to 79 years. Notably, multiple series consistently indicate that the mean age at presentation for sCRC tends to exceed that of spCRC in numerous cases [5].

Individuals diagnosed with inflammatory bowel diseases (IBDs), hereditary non-polyposis colorectal cancer (HNPCC), and familial adenomatous polyposis (FAP) are predisposed to an elevated risk of sCRC development. Specifically, sCRC constituted 2.5% of de novo CRC cases, 18% of ulcerative-colitis-related carcinoma cases, and 21% of FAP-related carcinoma cases, as reported by Nosho et al. [10].

### 1.2. Primary Management and the Prognosis of Synchronous Colorectal Cancer Compared to Solitary Primary Colorectal Cancer

Surgical resection stands as the principal therapeutic modality for sCRCs. A more comprehensive resection encompassing the removal of proximal intestinal segments and local lymph nodes in certain cases, particularly for patients with lesions located in adjacent segments, was advocated by Passman et al. [11].

However, the optimal surgical management of synchronous lesions in distinct colonic segments remains a subject of debate. Certain experts discuss total or subtotal colectomy, citing concerns over missed synchronous lesions during initial surgery. Such oversights may necessitate additional procedures, potentially leading to disease advancement and poorer prognostic outcomes for patients [9]. On the other hand, several researchers have proposed the effectiveness of extensive surgical interventions, such as proctocolectomy with J-pouch ileoanal anastomosis, total abdominal colectomy with ileorectal anastomosis, and proctosigmoidectomy with coloanal anastomosis [12].

The prognostic outlook for patients diagnosed with sCRC presents a spectrum ranging from better to equivalent or worse outcomes when compared to those with spCRC, as indicated in various studies. Such variance likely arises from discrepancies in sample sizes and durations of clinical monitoring, necessitating cautious interpretation.

A prospective investigation into the impact of sCRC on survival relative to spCRC was conducted by Nosho et al. in the United States in 2009 [13]. Their findings emphasized a significant association between sCRCs and a poorer prognosis, potentially attributed to the heightened risks of complications and metastasis inherent in multiple CRCs. However, in the other case cohorts, that survival outcomes for patients with sCRC were suggested to parallel those of individuals with spCRC [14]. The presence of sCRC alone may not wield sufficient predictive power as an independent determinant of survival rates.

## 2. The Relationship between Molecular Features and Colorectal Cancer Carcinogenesis

At present, three primary pathways are widely acknowledged to be implicated in the genesis of CRC: the chromosomal instability (CIN) pathway, the mutator-phenotype/DNA mismatch repair (MMR) pathway, and the hypermethylation phenotype hyperplastic/serrated polyp pathway (Figure 1) [15,16].

### 2.1. The Chromosomal Instability Pathway

The CIN pathway, exemplified by the inherited condition of FAP, is the predominant pathway in colorectal carcinogenesis. It is identified by changes in the number and structure of chromosomes and the buildup of somatic mutations in proto-oncogenes or genes related to tumors [17]. Structural changes, along with gains and losses of chromosome fragments, are distinctive traits of CIN tumors.

In CRC, the CIN pathway typically involves mutations and the loss of heterozygosity affecting tumor protein 53 (p53) and adenomatous polyposis coli (APC) [3]. These characteristics are potentially linked to higher mutation rates, with the majority of spCRCs falling into this group [18].

### 2.2. The Mutator-Phenotype/DNA Mismatch Repair Pathway

The mutator-phenotype/DNA mismatch repair pathway involves the loss of DNA mismatch repair protein function. MSI arises from a malfunction in the mismatch repair system, which causes changes in the number of repeated nucleotides within the coding exons. This results in frame-shift mutations in the associated genes. This pathway is associated with hereditary non-polyposis colorectal cancer, also known as Lynch syndrome, and some sporadic spCRCs in the elderly population. MSI has been identified in up to 15% of CRCs [19,20].

### 2.3. The Hypermethylation Phenotype Hyperplastic/Serrated Polyp Pathway

The hypermethylation phenotype hyperplastic/serrated polyp pathway is marked by a high frequency of methylation of certain CpG islands, termed CIMP-positive. In CIMP cases, tumor suppressor and DNA repair genes are frequently silenced transcriptionally. Epigenetic instability in this pathway is identified by the genome-wide simultaneous hypermethylation of promoter CpG island loci, leading to the inactivation of tumor-suppressor genes or genes related to tumors. The CIMP denotes a subset of CRCs that develop through this pathway of epigenetic instability [21,22].

Epigenetic alterations, such as DNA hypomethylation with the loss of imprinting and DNA hypermethylation, can silence the expression of specific genes, including those for MMR enzymes. The CIMP pathway encompasses the hypermethylation of the gene promoter for a DNA MMR gene. CIMP-positive tumors (a cytosine [C] base is immediately followed by a guanine [G] base linked with a phosphodiester bond [CpG]) are CRCs with a notably high frequency of methylation in certain CpG islands [23,24].

## 3. The Differences in Molecular Features and Related Pathways between Synchronous Colorectal Cancer and Solitary Primary Colorectal Cancer

Within a series derived from a general population of colorectal carcinoma cases, individuals with known predisposing conditions may contribute to slightly over 10% of sCRC occurrences [10]. Whether sCRC has unique prognostic and molecular features compared to spCRC remains uncertain. Molecular subtypes indicative of the CIN and MMR pathways did not exhibit concordance across lesions in individuals with sCRC [25]. These findings suggest that only a few cases exist where the mechanisms of carcinogenesis are fully understood in patients with sCRC.

### 3.1. MSI

The main theme of molecular biology research concerning sCRC focuses on the examination of MSI (Table 1). A noteworthy observation is the prevalence of MSI-positive colorectal carcinomas, which predominantly manifest as sporadic cases, attributed to methylation alterations in mismatch repair genes, rather than arising from germline mutations typical of HNPCC.

Investigations suggest that patients diagnosed with sCRC exhibit a heightened incidence of MSI-positive tumors compared to those with spCRC (Table 1). Additionally, analyses involving the methylation status of multiple genes or the presence of *BRAF* mutations in these malignancies indicate a sporadic origin for a considerable portion of sCRCs, rather than an inherited predisposition [14].

### 3.2. KRAS and BRAF

In addition to MSI, considerable attention has been directed towards investigating *KRAS* and p53 mutations in CRCs (Table 2). *KRAS* mutations serve as predictive indicators for the resistance to monoclonal antibody therapies targeting the EGFR, particularly in cases of metastatic colorectal carcinoma. Conversely, p53 mutations are frequently encountered in various human cancers and are often associated with more aggressive disease phenotypes. A notable observation across numerous studies is the presence of sCRC exhibiting discordant molecular profiles concerning MSI status, p53 mutation, and *KRAS* mutation within individual patients [33]. These findings explain the complex nature of sCRC formation, suggesting that its etiology is unlikely to be solely attributed to known genetic mutations typically implicated in CRC.

Given that anti-EGFR therapy is effective exclusively in individuals with *KRAS* or *BRAF* wild-type tumors, the wild-type status of *KRAS* and *BRAF* in the tumor assumes significance. An investigation revealing a low level of concordance between lesions, considering the potential clinical relevance for molecular targeted therapy, was mentioned by KEIICHI ARAKAWA et al. [25]. The concordance rates of *KRAS* and *BRAF* subtypes among cases of sCRC were observed to be less than 50%, indicative of a scenario where each lesion in sCRC may originate from distinct molecular pathways. 

### 3.3. CIMP

CIMP status was evaluated for subsets of CRCs based on the age of onset and the number of primary neoplasms: early-onset colorectal cancer (EOCRC) and late-onset colorectal cancer (LOCRC) with spCRCs, and sCRC by Ogino et al. [36]. The assessment of CIMP status showed that only 15.2% of EOCRC patients and 26.8% of LOCRC patients had CIMP-positive tumors [36]. However, 37.0% of sCRC patients exhibited CIMP positivity for both tumors and 27.8% of the tumor pairs showed at least one CIMP-positive tumor. These findings also suggest that the serrated pathway of carcinogenesis could be the primary mechanism of sCRC development [13,31].

There is evidence suggesting that sCRC might be more common among cancers following the serrated neoplasia pathway, marked by the CIMP phenotype, than among tumors of the CIN pathway (Table 2) [13,32]. sCRCs have been discovered to more commonly show the *BRAF c.1799T4A* mutation, which is a well-known indicator of the CIMP phenotype; these cases had a poorer outcome [13].

However, the relevance of the *BRAF* mutation for sCRC has been reported by only a few studies (Table 2) [25,32,34,35]. These results may be due to the diversity of CIMP-positive cancers, a subgroup that includes tumors that do not undergo *MLH1*-methylation, stay microsatellite-stable, and have a poor prognosis. In addition, *MLH1*-deficient MSI spCRC tends to have a much more favorable prognosis [35].

Varied methylation frequencies of individual CIMP markers suggest the presence of preferred epigenetic alterations in synchronous tumors, even when the overall CIMP phenotype remains unchanged. Methylation rates of *CACNA1G* and *NEUROG1* were higher for synchronous tumors than for solitary tumors, while that of *CDKN2A* (p16) was higher for solitary tumors than for synchronous tumors [35]. Intriguingly, the higher methylation rates of *CDKN2A* compared to those of other CIMP markers suggest potential favored epigenetic alterations in sCRC, even in cases where the overall CIMP phenotype appears unchanged [31].

## 4. Potential Prognostic Predictor Biomarker for the Therapeutic Strategy for sCRC

An improved understanding of the molecular mechanisms underlying the onset and progression of CRC may disclose novel therapeutic targets and biomarkers for risk assessment and stratification [37].

### 4.1. CIMP as a Potential Prognostic Predictor Biomarker

Individuals with CIMP-positive tumors exhibited especially poorer overall survival rates (HR = 2.06, 95%CI: 1.36–3.11, *p* = 0.0005) compared to those with CIMP-negative tumors in the 5-fluorouracil/leucovorin (FU/LV) arm, as reported by Muzny et al. [38]. Conversely, patients with CIMP-positive tumors demonstrated improved overall outcomes, with no significant disparity observed in comparison to CIMP-negative cases in the irinotecan, leucovorin, and fluorouracil (IFL) arm (HR = 0.90, 95%CI: 0.58–1.41, *p* = 0.65) [38]. Alternatively interpreted, individuals with CIMP-positive tumors trended towards enhanced overall survival when administered IFL versus FU/LV (69% vs. 56%, respectively; 95%CI: 0.37–1.05, *p* = 0.07). Among those with CIMP-negative tumors, IFL treatment yielded worse overall survival outcomes compared to FU/LV (68% vs. 78%; HR = 1.38, 95%CI: 1.00–1.89, *p* = 0.049) [38]. 

An increasing number of studies have proposed that CIMP could serve as a potential epigenetic predictor or prognostic biomarker, contributing to personalized and precise treatment for colorectal cancer [39,40]. When it comes to determining CIMP status, research commonly utilizes either the Ogino panel or the Weisenberger panel [41,42]. The Ogino panel incorporates eight markers, while the Weisenberger panel includes five markers, all of which are encompassed within the eight markers of the Ogino panel [41,42]. 

### 4.2. The Ogino Panel and the Weisenberger Panel for the Evaluation of CIMP Status

The Ogino panel, comprising all eight markers, demonstrated sensitivities exceeding 60% and specificities of 80% or higher for identifying CIMP-positive tumors [36]. This comprehensive panel proved notably superior in accurately reflecting the true CIMP status compared to alternative marker combinations, establishing it as the gold standard [36]. Ogino et al. affirmed that each of the eight markers serves as reliable surrogate indicators for determining CIMP status, with at least four markers—*RUNX3*, *CACNA1G*, *IGF2*, and *MLH1*—forming a particularly sensitive and specific CIMP panel [36]. On the other hand, the Weisenberger panel utilizes five markers: *CACNA1G*, *IGF2*, *NEUROG1*, *RUNX3*, and *SOCS1*, omitting *MLH1* from its selection [42]. Interestingly, despite these differences, the Weisenberger panel is more commonly employed compared to the Ogino panel. This preference may stem from the convenience of analyzing CIMP status with a smaller set of markers.

To assess CIMP, the methylation status of the promoter regions, *CACNA1G* (calcium channel, voltage-dependent, T type alpha-1G subunit), *CDKN2A* (cyclin-dependent kinase inhibitor 2A), *CRABP1* (cellular retinoic acid binding protein-1), *IGF2* (insulin-like growth factor 2), *MLH1*, *NEUROG1* (neurogenin-1), *RUNX3* (runt-related transcription factor 3), and *SOCS1* (suppressor of cytokine signaling 1), were examined [42]. Methylation of the 8-marker CIMP panel was categorized as CIMP-0 (none of the 8 markers was methylated), CIMP-low (1–4 of the 8 markers were methylated), and CIMP-high (≥5 markers were methylated). In addition, CIMP status was classified into a two-tiered system as CIMP-positive (CIMP-high) and CIMP-negative (CIMP-low plus CIMP-0) [42].

These findings align with the theory that sCRC originates from a single and isolated event, where different polyps evolve in parallel and transform simultaneously into colorectal cancer, primarily influenced by environmental factors rather than genetics [43].

## 5. Prediction of Therapeutic Drugs for CIMP-Positive sCRCs

The drugs taken into discussion included chemotherapy (5-Fluorouracil, irinotecan, oxaliplatin, and temozolamide), target therapy (cetuximab), DNA methylation inhibitor, and immunotherapy. Whether the drug is suggested to be beneficial or less effective to CIMP-positive CRC would be mentioned (Figure 2).

### 5.1. Chemotherapy with 5-Fluorouracil

#### 5.1.1. Adjuvant Chemotherapy with 5-Fluorouracil

5-Fluorouracil (5-FU) is the most commonly used chemotherapy for CRCs. It primarily functions as a thymidylate synthase inhibitor, disrupting DNA replication. However, an independent clinical trial of an Asian population yielded contrasting conclusions [44]. In the trial, 124 cases with stage II–III CRCs were included. The results revealed that the 3-year recurrence-free survival (RFS) of CIMP-positive CRC patients receiving a 5-FU-based regimen was significantly better (n = 17; RFS = 100%) than that of patients undergoing surgery alone (n = 7; RFS = 71.4%; *p* = 0.022) [44].

In another study, 103 stage III CRC cases with CIMP-positive status were treated with surgery alone and 103 cases were treated with surgery plus adjuvant 5-FU-based chemotherapy [45]. Patients with CRC manifesting a CIMP-positive profile experience significantly extended RFS when treated with surgical intervention combined with fluoropyrimidine-based adjuvant chemotherapy, compared to those undergoing solely surgical resection, despite the increased occurrence of the lymph node metastasis and advanced tumor-node-metastasis (TNM) stage within this group (*p* = 0.002) [45]. Moreover, this correlation was not significant for CIMP-negative CRC cases (*p* = 0.6); CIMP-positive status was posited as an independent and significant predictor of the survival benefit of CRC patients treated with adjuvant 5-FU-based chemotherapy [45]. These results align with prior research on the subject.

However, the CIMP status alone does not exert a significant influence on DFS among a CRC patient cohort [46]. Upon narrowing the focus to patients diagnosed with tumor-node-metastasis (TNM) stage II or III, CIMP status emerges as a crucial factor in determining the response to adjuvant chemotherapy involving 5-FU [46]. The results demonstrate that adjuvant 5-FU chemotherapy contributes to an improved DFS within the study cohort, with its efficacy closely tied to the CIMP status. Specifically, the benefits of adjuvant 5-FU therapy are distinctly observed in patients with CIMP-negative CRC [46]. Conversely, the administration of 5-FU-based adjuvant chemotherapy fails to yield enhanced DFS in individuals with CIMP-positive tumors [46]. Furthermore, among patients with TNM stage II or III CRC exhibiting CIMP positivity who opted out of adjuvant chemotherapy, significantly superior survival outcomes were observed [46].

The response of patients with CIMP subtypes to 5-FU-based chemotherapy on the basis of MSI, *BRAF*, and *KRAS* molecular classification was also reported by Murcia et al. [47]; 324 patients with stage II–III CRCs were categorized into six study subgroups [47]. The microsatellite-stable, *BRAF* wild-type, and CIMP-negative CRC subgroups showed longer disease-free survival when treated with 5-FU-based chemotherapy (log-rank test: *p* = 0.003 for the *KRAS* mutation subgroup and *p* < 0.001 for the *KRAS* wild-type subgroup) [47].

The critical role of intracellular folate concentrations in determining the efficacy of 5-FU treatment, particularly in CRCs exhibiting a CIMP-high status, have been highlighted by both Van Rijnsoever et al. and Iacopetta et al. [45,48]. These CIMP-high CRCs display elevated levels of 5–10-methylene tetrahydrofolate (CH2FH4) and tetrahydrofolic acid (FH4) compared to CIMP-low/negative CRCs, potentially explaining their enhanced response to 5-FU. 

#### 5.1.2. Chemotherapy with 5-Fluorouracil for Metastatic CRCs

Additionally, CIMP-high CRCs exhibit a significantly lower expression of γ-glutamyl hydrolase, an enzyme involved in reducing intracellular folate levels, which correlates with improved responses to 5-FU-based chemotherapy in metastatic CRC patients. Furthermore, the methylation-induced suppression of dihydropyrimidine dehydrogenase, a key enzyme in 5-FU degradation, may also contribute to the favorable response of CIMP-high CRCs to 5-FU.

### 5.2. Chemotherapy with Irinotecan

CIMP-positive status appears to be a potential biomarker for predicting the effectiveness of irinotecan-based chemotherapy for CRCs. However, this requires confirmation through further studies. Previous research indicated that demethylation treatment can activate multiple cancer cell signaling pathways. This not only permits the use of less toxic doses of irinotecan but also enhances its efficacy [49,50].

### 5.3. Chemotherapy with Oxaliplatin 

CIMP status may not be a significant prognostic biomarker for adjuvant oxaliplatin-based chemotherapy regimens in stage II–III CRCs. However, oxaliplatin tended to be less effective for CIMP-positive patients than for CIMP-negative patients [51].

### 5.4. Chemotherapy with Temozolomide

The methylation of the DNA repair gene O6-methylguanine-DNA methyltransferase promoter is a common and early occurrence in colorectal tumorigenesis, and it is believed to be beneficial for alkylating agents like temozolomide [52].

### 5.5. Target Therapy

Cetuximab is a widely used monoclonal antibody (IgG1) that specifically targets the overexpressed epidermal growth factor receptor (EGFR) in metastatic CRCs. A Japanese study found that the progression-free survival (PFS) of patients with CIMP-positive and *KRAS* wild-type CRCs treated with cetuximab was shorter than that of patients with CIMP-negative and *KRAS* wild-type CRCs (median PFS of 2.1 vs. 5.1 months, *p* = 0.11). In addition, the objective response rate was lower (20.0 vs. 24.4%, *p* = 0.90) [53]. While its results were not statistically significant, the study suggested that the CIMP-positive phenotype might act as a negative biological predictor of the effectiveness of the anti-EGFR antibody [53]. Furthermore, cetuximab against CIMP-positive stage III CRCs showed a non-significant trend of negative efficacy [54].

### 5.6. DNA Methylation Inhibitor

DNMT1, DNMT3A, and DNMT3B are the main cytosine-5 DNA methyltransferase enzymes. They play a role not only in creating and preserving DNA methylation patterns but also in regulating various gene functions. This includes activities such as transcriptional silencing, transcriptional activation, and posttranscriptional regulation [55]. 

The most extensively studied DNA methylation inhibitors include 5-azacytidine (azacitidine or 5-aza-CR), 5-aza-2′-deoxycytidine (decitabine or 5-aza-CdR), and guadecitabine (SGI-110). These inhibitors form irreversible complexes with DNMTs by replacing methylated cytosine targets during DNA replication. This process leads to the depletion of the enzyme and cytosine during cell division, resulting in passive DNA demethylation. As a result, tumor suppressor genes are re-expressed, proliferation is controlled, and the carcinogenesis process is inactivated [56,57].

It has been suggested that CIMP-positive CRCs might potentially benefit from treatment with DNA methylation inhibitors alone or in combination [26,27]. In phase I/II clinical trials, sequential decitabine and panitumumab were well-tolerated by 20 patients with *KRAS* wild-type metastatic CRCs [58].

While low-dose DNA methylation inhibitors can cause demethylation and promote apoptosis, *DNMT* inhibition alone might not be enough to achieve the lasting and strong re-expression of transcriptional genes. Therefore, DNA methylation inhibitors are considered as immune modulators that induce *CTA* expression in CRCs, triggering cytotoxic T-cell responses and promoting antitumor immunity [59]. The combining of anti-EGFR antibodies or immunotherapy to DNA methylation inhibitors seems promising.

Epigenetic therapies encompass not only DNA methylation inhibitors but also *HDAC* inhibitors, *BET* inhibitors, and *EZH2* inhibitors. Different types of epigenetic alterations require specific interventions. The advancement of these therapies holds promising opportunities for innovative and enhanced therapeutic interventions for CRCs [60].

### 5.7. Immunotherapy

While no direct correlation has been established between CIMP status and distinct levels of tumor-infiltrating lymphocytes (TILs) such as forkhead box protein P (FOXP3) or cluster of differentiation 8 (CD8), the observed association of CIMP status with TILs indicates the potential involvement of the immune system in CIMP-positive CRC [61]. Further investigation, particularly considering MSI status, is warranted to emphasize this relationship.

This character positions CIMP-positive tumors as promising candidates for immunotherapy investigations, including trials involving anti-programmed cell death protein 1 (anti-PD1) therapy [62]. Notably, CIMP-positive tumors have been observed to correlate with the presence of programmed cell death protein 1 (PD-1)-positive T cells in a subset of MSI-high cancers, enhancing the potential relevance of targeting CIMP-positive tumors in immunotherapeutic strategies [62]. CIMP-low tumors were characterized to be linked with tumor immune microenvironment type 2 (TIME-2) and tumor immune microenvironment type 3 (TIME-3) subtypes, distinguished by the presence of TILs [63].

Likewise, the presence of the Crohn-like infiltrate signifies an association between CIMP status and peritumoral lymphoid aggregates, underscoring the involvement of the immune system in both MSI-high and CIMP-positive cancers [64]. Crohn-like infiltrates in CRC are associated with a favorable prognosis, suggesting the involvement of the host immune response in impeding cancer progression. This phenomenon may represent a mechanism influencing prognosis in CIMP-positive tumors.

## 6. Conclusions

Synchronous colorectal tumors often show MSI, CIMP, and *BRAF* mutations [13]. In particular, sCRCs have been recognized to be CIMP-positive at a higher frequency than spCRCs. A tendency to high MSI rates has also been reported [65]. Recent systematic reviews have confirmed the association between the CIMP phenotype and certain demographic and clinical features, including older age, female gender, proximal tumor location, mucinous histology, poor differentiation, and MSI [66]. 

Colorectal cancer provides a compelling model for studying the molecular pathogenesis of cancer. Colorectal tissue is easily accessible for biopsy, and there is a clear progression from normal colonic epithelium to invasive cancer through an intermediate precursor, the adenomatous polyp. sCRCs, which occur in the same person due to common causes, offer a unique model to study how three different molecular pathways contribute to the development of individual tumors. 

CpG-island methylation testing has been recommended as a tool for cancer detection and prognostication, emphasizing the clinical relevance of the methylation status [67]. However, the timing of methylation and its interaction with other genetic defects are not fully understood [28,30,68].

A significant challenge in CIMP research lies in the absence of a universally accepted standard definition for assessing and quantifying CIMP status in patients with CRC [29]. Moreover, numerous features associated with CIMP-positive tumors overlap with those observed in high MSI-high cancers. Subsequent investigations should endeavor to evaluate these correlations while stratifying by MSI status. This approach will enable a comprehensive examination of the molecular pathways influencing CIMP-positive tumors, independent of their MSI status.

The studies in this review were limited because molecular analysis was not performed for all tumors in patients with sCRCs who met the inclusion criteria. In addition, patients were chosen from hospital-based samples; therefore, the chance of selection bias in these samples, compared to population-based samples, cannot be dismissed. In future studies, it is worth considering how immunotherapy and DNA methylation inhibitors can work together.

## Figures and Tables

**Figure 1 ijms-25-05243-f001:**
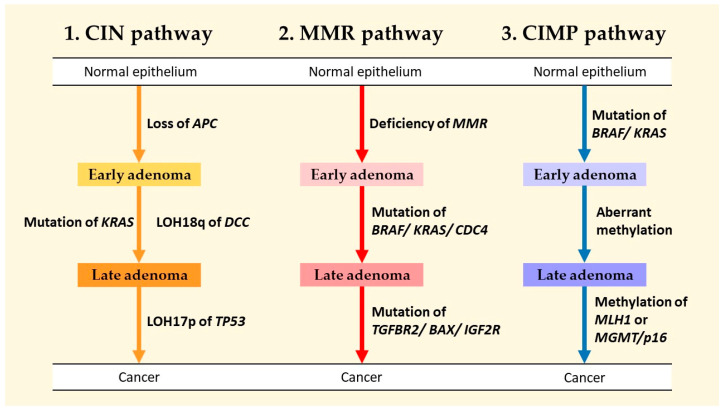
Illustration showing the chromosomal instability (CIN) pathway, the mutator-phenotype/DNA mismatch repair (MMR) pathway, and the hypermethylation phenotype hyperplastic (CIMP)/serrated polyp pathway, and their relationship between molecular features and CRC carcinogenesis. (*APC*: adenomatous polyposis coli gene. *KRAS*: Kirsten rat sarcoma viral oncogene homologue. LOH18q: loss of heterozygosity on chromosome 18q. DCC: deleted in colorectal cancer gene. LOH17p: loss of heterozygosity on chromosome 17p. *BRAF*: B-Raf proto-oncogene serine/threonine kinase gene. CDC4: cell division control protein 4. TGFBR2: transforming growth factor, beta receptor II. BAX: bcl-2-like protein 4. IGF2R: insulin like growth factor 2 receptor. *MLH1*: MutL protein homolog 1. MGMT: O6-methylguanine-DNA methyltransferase. p16: p16 protein.).

**Figure 2 ijms-25-05243-f002:**
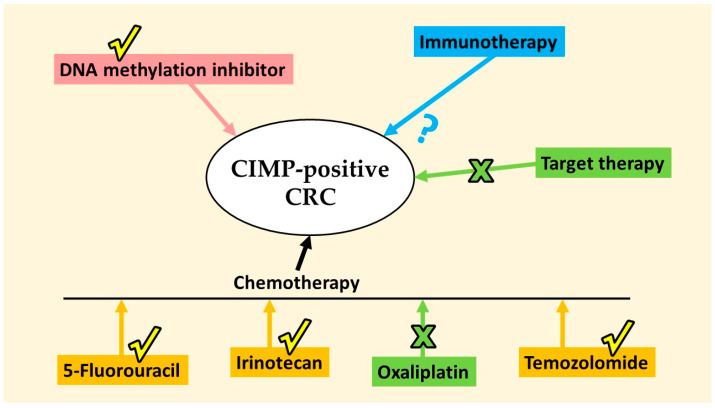
Prediction of therapeutic drugs for CIMP-positive CRCs. The check mark indicated the possible benefit to the population, while the cross mark indicated evidence being of less effective to the group. The question mark pointed out that future investigation is warranted for immunotherapy use.

**Table 1 ijms-25-05243-t001:** Review articles of the correlations between CRCs and molecular features.

Research Topic	Types	Year	Key Findings	Reference
DNA methylation aberrancies delineate clinically distinct subsets of CRC and provide novel targets for epigenetic therapies	LR ^2^	2018	Information on DNA methylation alterations in CRC, the importance of CIMP status, specific molecular profiles, and the application of epigenetic therapies for CRC.	[26]
MSI in CRC	LR ^2^	2018	Summary of the CRC classification and diagnostic features of MSI.	[19]
Aberrant DNA Methylation in CRC: What Should We Target?	LR ^2^	2017	Summary of clinical trials of DNA methyltransferase inhibitors in CRC	[27]
Serrated CRC: Molecular classification, prognosis, and response to chemotherapy	LR ^2^	2016	Summary the serrated pathway of CRC, including CRC molecular and clinical features, prognosis, and response to chemotherapy.	[28]
Tracking the Correlation Between CIMP and Other Molecular Features and Clinicopathological Features in Human CRCs	SR ^3^ & MA ^4^	2016	Suggesting CIMP could be used as an independent prognostic marker for CRCs.	[29]
Serrated Polyps and Their Alternative Pathway to the CRC	SR ^3^	2015	The SSA/P ^1^ subtypes of serrated polyps are the precursors of CRC by MSI.	[30]
Serrated polyps and the risk of synchronous colorectal advanced neoplasia	SR ^3^ & MA ^4^	2015	Serrated polyps are associated with a more than two-fold increased risk of detection of synchronous advanced neoplasia.	[31]
Multiple Sporadic CRCs Display a Unique Methylation Phenotype	LR ^2^	2014	Multiple CRC are associated with a distinct methylation phenotype, with a close association between tumor multiplicity and CIMP-high.	[32]
sCRC: clinical, pathological and molecular implications	LR ^2^	2014	Clinical, pathological, and molecular characteristics of sCRC as compared to spCRC.	[5]
Molecular classification and correlates in CRC	LR ^2^	2008	Molecular classification and molecular correlates based on MSI status and CIMP status in CRC.	[16]

CRC: synchronous colorectal cancer. sCRC: synchronous colorectal cancer. CRC: colorectal cancer. MSI: microsatellite stabilities. CIMP: CpG island methylator phenotype. ^1^ SSA/p: sessile serrated adenoma/polyp. spCRC: sporadic primary colorectal cancer. ^2^ LR: literature review. ^3^ SR: systematic review. ^4^ MA: meta-analysis.

**Table 2 ijms-25-05243-t002:** Different prevalence reports of aberrant molecules/pathways including MSI, CIMP, *BRAF*, and *KRAS* in sCRC.

Research Topic	Year	Model	MSI	CIMP	*BRAF*	*KRAS*	Reference
CpG island methylator phenotype is associated with response to adjuvant irinotecan-based therapy for stage III colon cancer	2019	In vivo	*p* = 0.3453 (NS)	-	*p* = 0.8884 (NS)	*p* = 0.2033 (NS)	[25]
Multiple sporadic colorectal cancers display a unique methylation phenotype	2014	In vivo	*p* = 0.4 (NS)	*p* = 0.004 (SS)	*p* = 1 (NS)	*p* = 1 (NS)	[32]
Molecular heterogeneity and prognostic implications of synchronous advanced colorectal neoplasia	2014	In vivo	*p* = 0.64 (NS)	-	*p* = 0.96 (NS)	-	[34]
Clinicopathologic and molecular characteristics of synchronous colorectal cancers: heterogeneity of clinical outcome depending on microsatellite instability status of individual tumors	2012	retrospective	*p* < 0.001 (SS)	*p* = 0.14 (NS)	*p* > 0.99 (NS)	*p* = 0.83 (NS)	[35]
A prospective cohort study shows unique epigenetic, genetic, and prognostic features of synchronous colorectal cancers	2010	In vivo	*p* = 0.037 (SS)	*p* = 0.013 (SS)	-	-	[13]
			**MSI + CIMP**				
			*p* = 0.036 (SS)				

SS: statistically significant. NS: not significant. MSI: microsatellite instability; CIMP: CpG island methylator phenotype; *BRAF*: B-Raf proto-oncogene serine/threonine kinase; *KRAS*: Kirsten rat sarcoma viral oncogene homologue.

## Data Availability

All the data are included in manuscript.

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
