# Peer review of "The CpG Island Methylator Phenotype Status in Synchronous and Solitary Primary Colorectal Cancers: Prognosis and Effective Therapeutic Drug Prediction"

_ijms, 2024, doi:10.3390/ijms25105243_

Round 1

Reviewer 1 Report

Comments and Suggestions for Authors

This review article is well written. However, some points described below would be better to revise.

Major comments

1.     Outline, section, and description seem a little scattered throughout the manuscript. The characteristics and managements for synchronous CRC should be summarized in introduction. Section 5 is probably the most important section including Table 1 and 2, which are only Tables and Figures presented in this manuscript. However, this section is somewhat difficult to understand. Therapeutic strategy should be discussed to separate adjuvant and metastatic chemotherapy in Sections 6 and 7.

2.     Please make the manuscript more attractive for the readership using graphic figures and summary tables.

Minor comments

1.     Please check the abbreviations first to go out.

2.     Table 1 merely shows the molecular research review papers on MSI including sCRC. Is this informative and really necessary?

3.     Table 2 seems pivotal in this manuscript, but is not so readable. What means as “variation” in the table title and “p (differences of what)” in the table. This table mainly referred to BRAF in the text. Please modify the table to be understandable and describe more other than BRAF in the text.

4.     Please make Section 7.1 short, since the use of 5-FU alone is less common in patients with metastatic CRC other than the elderly.

Reviewer 2 Report

Comments and Suggestions for Authors

This review manuscript has a scientific interest that might benefit the readers, but some revisions still need to be included. Therefore, I recommend that the authors modify the review manuscript.

1.      The authors should add more detail about this review study's aims, objectives, and clinical significance at the end of the Introduction part.

2.      The authors are suggested to make a schematic illustration in section 1. Introduction showing that sCRC might be more common in the serrated neoplasia pathway, marked by the CpG island methylator phenotype 15 (CIMP), than in the chromosomal instability pathway mentioned in the abstract.
Figure 1: sCRC might be more common in the serrated neoplasia pathway, marked by the CpG island methylator phenotype 15 (CIMP)

3.      The authors must make a schematic illustration in section 4, The Relationship Between Molecular Features and Colorectal Cancer Carcinogenesis, which shows all the relationships between the three mentioned pathways for better understanding by the readers. These pathways include the Chromosomal Instability Pathway, The Mutator-Phenotype/DNA Mismatch Repair Pathway, The Hypermethylation Phenotype Hyperplastic/Serrated Polyp Pathway, and their relationship between molecular features and colorectal cancer carcinogenesis.
Figure 2:
Illustration showing Chromosomal Instability Pathway, The Mutator-Phenotype/DNA Mismatch Repair Pathway, The Hypermethylation Phenotype Hyperplastic/Serrated Polyp Pathway, and their relationship between molecular features and colorectal cancer carcinogenesis

4.      The authors must make a schematic illustration in section 7, Prediction of Therapeutic Drugs for CIMP-Positive sCRCs, illustrating all the therapy drug options and demonstrations mentioned in the written form in the review manuscript text.
Figure 3: Prediction of Therapeutic Drugs for CIMP-Positive sCRCs, chemotherapy 5-Fluorouracil, Irinotecan, Oxaliplatin, Temozolomide, DNA Methylation Inhibitor and Immunotherapy

5.      The authors must add more literature surveys in Table 1, especially research articles showing the in-vivo mechanism of sCRC cancer. Also, in Table 1, the authors must add the study name in the column instead of the authors, which makes the table more descriptive. This version of Table 1 needs to be more scientifically appealing, and there is no scientific significance per the mentioned review topic. We suggest revising Table 1 fully, adding more columns, variables, and a more in-depth literature survey. Authors can add in-vitro and in-vivo studies to support their research point.

 Authors can refer to the review article below for better understanding, as mentioned in Table 1. Farooqi, M.A.; Mahnoor, M.; Delgado, K.M.; Dahlgren, W.T.-T.; Kang, C.-U.; Farooqi, H.M.U. Focused Ultrasound as Targeted Therapy for Colorectal Cancer: A Comprehensive Review. Gastrointest. Disord. 2024, 6, 380-401. https://doi.org/10.3390/gidisord6020026.

6.      The authors should specify the type of article and study name in the Study column in Table 2, The variation in molecular features of synchronous colorectal cancers. They must also add more data in TABLE 2 to make the review article more comprehensive.

Round 2

Reviewer 1 Report

Comments and Suggestions for Authors

The title of Table 2 still seems difficult to understand. It may be better to simplify as follows: Different prevalence reports of aberrant molecules/pathways including MSI, CIPM, BRAF, and KRAS in sCRC.

Reviewer 2 Report

Comments and Suggestions for Authors

After the incorporation of suggestions and revisions, this review manuscript now appears more research-oriented, with systematic and schematic illustrations that make it more valuable for the scientific reader.

Best wishes.
